# R1-SyntheticVL: Is Synthetic Data from Generative Models Ready for Multimodal Large Language Model?

**Jingyi Zhang** [1 2 *]  **Tianyi Lin** [1 *]  **Huanjin Yao** [3]  **Xiang Lan** [4]  **Shunyu Liu** [2]  **Jiaxing Huang** [1]

## Abstract

In this work, we aim to develop effective data synthesis techniques that autonomously synthesize multimodal training data for enhancing MLLMs in solving complex real-world tasks. To this end, we propose Collective Adversarial Data Synthesis (CADS), a novel and general approach to synthesize high-quality, diverse and challenging multimodal data for MLLMs. The core idea of CADS is to leverage collective intelligence to ensure high-quality and diverse generation, while exploring adversarial learning to synthesize challenging samples for effectively driving model improvement. Specifically, CADS operates with two cyclic phases, i.e., Collective Adversarial Data Generation (CAD-Generate) and Collective Adversarial Data Judgment (CAD-Judge). CAD-Generate leverages collective knowledge to jointly generate new and diverse multimodal data, while CAD-Judge collaboratively assesses the quality of synthesized data. In addition, CADS introduces an Adversarial Context Optimization mechanism to optimize the generation context to encourage challenging and high-value data generation. With CADS, we construct MMSynthetic-20K and train our model R1-SyntheticVL, which demonstrates superior performance on various benchmarks. Code, model and data will be available at https://github.com/jingyi0000/R1-SyntheticVL.

## 1. Introduction

Recent advances in Multimodal Large Language Models (MLLMs) (Achiam et al., 2023; Bai et al., 2023; Tong et al., 2024; Team et al., 2023) have demonstrated remarkable progress, enabling AI systems to perceive, understand and reason across diverse modalities. The great success of MLLMs, however, highly relies on the availability of large-scale multimodal training data. Unfortunately, AI is running out of data (Jones, 2024), especially for domain-specific data (e.g., medical and safety-sensitive data) that are inherently scarce and hard to obtain. On the other hand, even the raw data is available, annotating large-scale multimodal data is prohibitively expensive and time-consuming (Yao et al., 2024a; Xu et al.), particularly for complex real-world tasks that require reasoning and Chain-of-Thought (CoT) annotations. These challenges significantly hinder the continued development of MLLMs.

Data synthesis has emerged as a promising alternative to alleviate data constraints in LLMs (Wang et al., 2023; Ding et al., 2025; Li et al., 2024b; Luo et al., 2023). By leveraging high-quality synthetic training data, recent attempts have demonstrated their effectiveness in enhancing LLMs' instruction-following and reasoning capabilities, significantly reducing the reliance on real data. For example, SELF-INSTRUCT (Wang et al., 2023) generates new instruction-following data automatically while Scale-Quest (Ding et al., 2025) synthesizes effective large-scale mathematical reasoning data. Inspired by the success of data synthesis in LLMs, a natural question arises: *is synthetic data from generative models ready for MLLMs?*

Unlike LLMs, synthesizing multimodal data requires generating high-quality visual content, which has been a long-standing challenge, especially for complex real-world tasks that involve fine-grained visual details, such as precise spatial relationships, rigorous factuality, etc. Recent breakthroughs in text-to-image generation, represented by Nano Banana Pro (DeepMind, 2025b), demonstrate unprecedented capabilities in visual generation, opening a promising way for multimodal data synthesis.

However, directly applying Nano Banana Pro often suffers from several issues as illustrated in Fig. 1: (1) Data Quality: the generated QA data may exhibit multimodal misalignment and factual inconsistencies (e.g., visual evidence like unclosed shapes in the question image that contradicts the question text and answer), leading to low-quality multi-

---

*Equal contribution   [1]Hong Kong Polytechnic University [2]Nanyang Technological University [3]Tsinghua University [4]National University of Singapore. Correspondence to: Jiaxing Huang <jiaxing.huang@polyu.edu.hk>.

*Proceedings of the 43rd International Conference on Machine Learning*, Seoul, South Korea. PMLR 306, 2026. Copyright 2026 by the author(s).

## (a) Directly using Nano Banana Pro

**Data Quality:** Misalignment

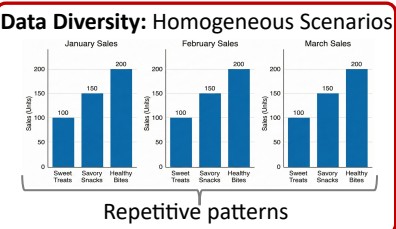

**Q:** Given a right triangle ABC, *angle C is 90°, AC=CD* ...

**Data Diversity:** Homogeneous Scenarios

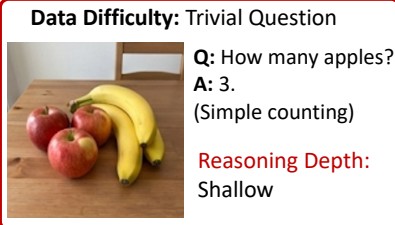

Repetitive patterns

**Data Difficulty:** Trivial Question

**Q:** How many apples?
**A:** 3.
(Simple counting)

Reasoning Depth: Shallow

## (b) Our proposed Collective Adversarial Data Synthesis (CADS)

**Data Quality:** Precise alignment

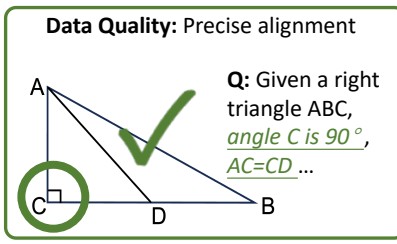

**Q:** Given a right triangle ABC, *angle C is 90°, AC=CD* ...

**Data Diversity:** Rich Scenarios

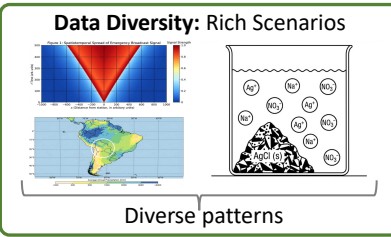

Diverse patterns

**Data Difficulty:** Complex Reasoning

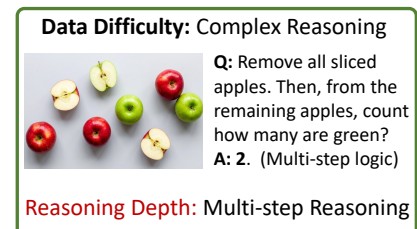

**Q:** Remove all sliced apples. Then, from the remaining apples, count how many are green?
**A: 2.** (Multi-step logic)

Reasoning Depth: Multi-step Reasoning

*Figure 1.* **Comparison between "directly using Nano Banana Pro" and our proposed "Collective Adversarial Data Synthesis (CADS)".** (a) Directly applying Nano Banana Pro often suffers from low data quality, limited data diversity and trivial difficulty in data synthesis of complex tasks, while (b) our proposed CADS effectively synthesizes high-quality, diverse and challenging multimodal data for MLLMs.

modal data; (2) Data Diversity: due to the inherent model biases, the generated data often suffers repetition and homogeneity; and (3) Data Difficulty: it is hard to generate high-quality non-trivial data (e.g., yielding challenging questions that are informative and useful to model training instead of easy questions that are generally trivial), thereby failing to synthesize useful training data.

To address these issues, we propose Collective Adversarial Data Synthesis (CADS), a novel and general approach to synthesize high-quality, diverse and challenging multimodal data for MLLMs, enhancing the capabilities in complex real-world tasks. The core idea of CADS is to leverage collective intelligence to ensure high-quality and diverse generation, while exploring adversarial learning to synthesize challenging samples for effectively driving model improvement. Specifically, CADS operates with two cyclic phases, including Collective Adversarial Data Generation (CAD-Generate) and Collective Adversarial Data Judgment (CAD-Judge). In the generation phase, CAD-Generate leverages collective knowledge from multiple MLLMs to jointly generate new and diverse multimodal data (i.e., question text and image, and answer). In the judgment phase, CAD-Judge employs multiple MLLMs as judges to collaboratively assess the quality of synthesized data, enabling to identify and filter out low-quality instances with issues like semantic misalignment or factual errors. During the generation and judgment cycle, CADS introduces an Adversarial Context Optimization mechanism, which identifies adversarial instances that are high-value and challenging, and

then employs them to optimize the generation context to dynamically refine the data synthesis heuristics, encouraging challenging and high-value data generation.

In this way, our proposed CADS enables high-quality, diverse and challenging multimodal data synthesis. (1) The collective judgment mechanism ensures data quality. By aggregating verdicts from multiple MLLM judges, CADS effectively identifies and filters out the samples with multimodal misalignment and factual inconsistencies, guaranteeing the reliability of the synthesized data; (2) The collective generation strategy encourages data diversity. Leveraging the collective intelligence of multiple MLLMs allows CADS to break free from the inherent biases and repetitive generation of a single model, enriching the generated data with varied perspectives; (3) The adversarial context optimization mechanism guarantees sufficient difficulty. By optimizing the generation context using high-value adversarial instances, CADS forces the synthesis beyond trivial patterns to generate challenging samples, offering useful information for model improvement.

Based on our CADS, we synthesize MMSynthetic-20K, a high-quality, diverse and challenging multimodal dataset, which provides a valuable resource comparable to real data for advancing MLLM research. With MMSynthetic-20K, we perform reinforcement learning (i.e., GRPO) on it to train our model, R1-SyntheticVL, which demonstrates superior capabilities in handling various complex real-world multimodal tasks.

The main contributions of this work are summarized as follows: First, we propose Collective Adversarial Data Synthesis (CADS), a novel and general approach to synthesize high-quality, diverse and challenging multimodal data for MLLMs. To the best of our knowledge, this is the first work that explores synthetic multimodal data from generative models for MLLMs and introduces the idea of collective adversarial learning into MLLM data synthesis. Second, based on CADS, we construct MMSynthetic-20K, a high-quality, diverse, and challenging multimodal dataset that provides a valuable resource for mitigating data scarcity and advancing research in MLLM training. Third, we develop R1-SyntheticVL, a powerful MLLM trained via Reinforcement Learning (i.e., GRPO) using our synthetic data, demonstrating exceptional capabilities in handling complex real-world multimodal tasks. Fourth, extensive experiments demonstrate the superiority of our proposed method and model on various benchmarks, validating the effectiveness of our data synthesis framework.

## 2. Related Work

### 2.1. Multimodal Large Language Model

Multimodal Large Language Models (MLLMs) (Achiam et al., 2023; Bai et al., 2023; Yang et al., 2024; Tong et al., 2024; Wu et al., 2025; Chen et al., 2024; Lu et al., 2025; Team et al., 2023; Yao et al., 2024d;b;a; 2025c; Zhang et al., 2024a; Huang et al., 2025a; Yao et al., 2026) have achieved significant progress in diverse vision-language tasks, demonstrating exceptional capabilities in interpreting and analyzing visual information across various application domains. Early works on MLLMs mainly center on image-text alignment and modality integration (Achiam et al., 2023; Bai et al., 2023; Wu et al., 2025; Lan et al., 2025; Jin et al., 2025), while recent advancements have shifted towards incentivizing the reasoning capabilities of MLLMs to address complex real-world problems (Yao et al., 2024a; Zhang et al., 2023; Xu et al.; Zhang et al., 2025a; Yang et al., 2025a; Peng et al., 2025; Zhan et al., 2025; Yaowei et al., 2025; Yao et al., 2025b; Ma et al., 2026). To achieve this, representative works like LLaVa-CoT (Xu et al.) and Mulberry (Yao et al., 2024a) leverage strong teacher models (e.g., GPT-4 (OpenAI, 2024)) to generate high-quality Chain-of-Thought (CoT) data for supervised fine-tuning. Furthermore, driven by the success of reinforcement learning, emerging studies have also demonstrated promising potential in enhancing MLLM reasoning through actively self-exploration. Different from previous studies, we explore data synthesis for MLLMs and propose Collective Adversarial Data Synthesis (CADS), which effectively synthesizes high-quality, diverse and challenging multimodal data for MLLMs.

### 2.2. Data Synthesis

Data synthesis has proven to be a promising way for alleviating data constraints in deep learning. In the field of image recognition (He et al., 2022), generative models such as GANs (Zhu et al., 2017; Goodfellow et al., 2020), GLIDE (Nichol et al., 2021), diffusion models (Ho et al., 2020; Rombach et al., 2022) are employed to generate synthetic data targeted for a specific label space, bringing performance boosts for image recognition tasks. In the era of large language models, data synthesis has become even more important due to the demand for large volumes of high-quality training data (Wang et al., 2023; Lu et al., 2024; Ding et al., 2025; Seegmiller et al., 2025; Qin et al., 2025; Li et al., 2024b; Wang et al., 2025a; Huang et al., 2024; Luo et al., 2023; Xu et al., 2024). For instance, SELF-INSTRUCT (Wang et al., 2023) improves the instruction-following capabilities of LLMs using their own generated new instruction data. ScaleQuest (Ding et al., 2025) introduces a cost-effective data synthesis method to generate large-scale mathematical reasoning data.

However, data synthesis is largely under-explored for MLLMs. Due to the limitations in generating high-quality visual content, existing attempts (Yang et al., 2025b; Huang et al., 2025b; Shi et al., 2025; Linger et al., 2025; Zhang et al., 2025b; Zhao et al., 2024) generally focus solely on synthesizing the textual modality for existing images, or rely on programmatic engines (e.g., Python plotting package) to generate specific visual structures. For example, Oasis (Zhang et al., 2025b) employs MLLMs to generate diverse instructions for given images, significantly enhancing the performance of MLLM with various backbones. ECD (Yang et al., 2025b) improves the MLLMs' capabilities in chart understanding by synthesizing chart data using Python's chart plotting packages. TR-CoT (Linger et al., 2025) enhances the geometric reasoning for MLLMs by synthesizing theorem-grounded geometric diagrams, which are constructed by integrating various geometric substrates with underlying theorems. Different from previous works, we explore generative models (i.e., Nano Banana Pro (DeepMind, 2025b)) for multimodal data synthesis and propose a general approach for generating high-quality, diverse and challenging data, effectively enhancing the capabilities of MLLMs in complex real-world tasks.

## 3. Method

We first provide the analysis of our motivation in Sec. 3.1, and then present our proposed Collective Adversarial Data Synthesis (CADS) in Sec. 3.2. More details to be elaborated in the ensuing subsections.

*Table 1.* **Comparison between *Directly Applying* SOTA generative models and our proposed CADS on multimodal data synthesis.**

| Benchmark | Qwen2.5-VL-7B | Stable Diffusion | Nano Banana | Nano Banana Pro | CADS (Ours) |
|-----------|---------------|------------------|-------------|-----------------|-------------|
| MathVista | 68.2 | 66.3 | 67.9 | 70.8 | **75.6** |

## 3.1. Analysis and Motivation

As discussed, the scarcity of high-quality data largely hinders the continued development of MLLMs, making automated data synthesis an essential alternative. This section analyzes the challenges in multimodal data synthesis. Specifically, we first explain why previous text-to-image generation methods fail in high-quality multimodal data synthesis and how Nano Banana Pro offers a promising way. In addition, we demonstrate that directly applying Nano Banana Pro exhibits limitations across several aspects. We support our analysis with illustrations and experiments.

**Challenges in multimodal data synthesis.** Compared to data synthesis in LLMs, multimodal data synthesis for MLLMs requires generating high-quality visual content that aligns strictly with complex textual content. While previous text-to-image models such as Stable Diffusion (Rombach et al., 2022) and Nano Banana (DeepMind, 2025a) show strong capabilities in image generation, they often fall short when applied to complex real-world tasks requiring fine-grained visual details, such as precise spatial relationships, rigorous factuality. By contrast, the recent breakthrough, Nano Banana Pro (DeepMind, 2025b), demonstrates more powerful image generation capabilities, particularly in handling complex real-world tasks. As shown in Table 1, Stable Diffusion and Nano Banana degrade performance compared with the baseline, highlighting their inadequacy for generating high-fidelity training data. Conversely, Nano Banana Pro brings performance improvement, verifying its feasibility for effective multimodal data synthesis.

**Limitations of directly applying Nano Banana Pro.** Despite the strong capability of Nano Banana Pro, our analysis shows that directly applying Nano Banana Pro for autonomous data synthesis remains suboptimal. As illustrated in Fig. 1, directly applying Nano Banana Pro suffers from three critical bottlenecks, including low data quality, limited data diversity and data difficulty. To further support our observation, we conduct an experiment comparing a model trained on data synthesized by directly using Nano Banana Pro against our proposed CADS. As shown in Table 1, directly applying Nano Banana Pro yields marginal performance gains, significantly underperforming compared to our method. This further demonstrates that a base generator alone is insufficient for generating high-quality multimodal data, and a more sophisticated framework is required to ensure quality, enforce diversity, and encourage challenging data for model training.

## 3.2. Collective Adversarial Data Synthesis

We propose Collective Adversarial Data Synthesis (CADS), a novel and general approach to synthesize high-quality, diverse and challenging multimodal data for MLLMs, enhancing the capabilities of MLLMs in tackling complex real-world tasks. As illustrated in Fig. 2, CADS operates with two cyclic phases, including Collective Adversarial Data Generation (CAD-Generate) and Collective Adversarial Data Judgment (CAD-Judge). During these two phases, CADS introduces an Adversarial Context Optimization mechanism to further encourage challenging and high-value data generation.

### 3.2.1. COLLECTIVE ADVERSARIAL DATA GENERATION

Collective Adversarial Data Generation (CAD-Generate) leverages collective knowledge from multiple MLLMs to jointly generate new and diverse multimodal data. CAD-Generate starts with a set of seed data $\mathcal{D}_{seed}$, where each seed data $D \in \mathcal{D}_{seed}$ can be either a multimodal sample or a textual description of a target task, allowing our CADS to generate new data either from existing data or from scratch. Then, we employ a group of MLLMs (i.e., $\{\pi_1, \pi_2, ..., \pi_K\}$) to jointly generate new synthetic multimodal data (i.e., $\mathcal{D}_{syn} = \{(v'_j, q'_j, a'_j)\}_{j=1}^M$):

$$\mathcal{D}_{syn} = \mathcal{G}(D), \tag{1}$$

where the generation process $\mathcal{G}$ consists of three main steps.

**Step 1: Rationale analysis.** This step identifies the core knowledge concepts and analyzes the underlying reasoning logic of the seed data. Specifically, given a seed data $D \in \mathcal{D}_{seed}$, each MLLM $\pi$ is employed to extract its knowledge domain (e.g., Geometry, Physics, Biology) and the underlying rationale required to solve the problem, which are used as the semantic contexts for the subsequent generation phase. In this way, we can ensure that the newly synthesized data strictly adhere to a valid rationale and reasoning logic, thus significantly guaranteeing the quality of the generated data.

**Step 2: Synthesis strategy generation.** Based on the extracted domain information and rationale from the previous step, we generate a specific strategy to synthesize the new problem, i.e., $(q', a')$. Specifically, we define four meta strategies:

- *(1) Numerical & Parameter Variation:* it maintains the seed data's structure and reasoning logic but systemati-

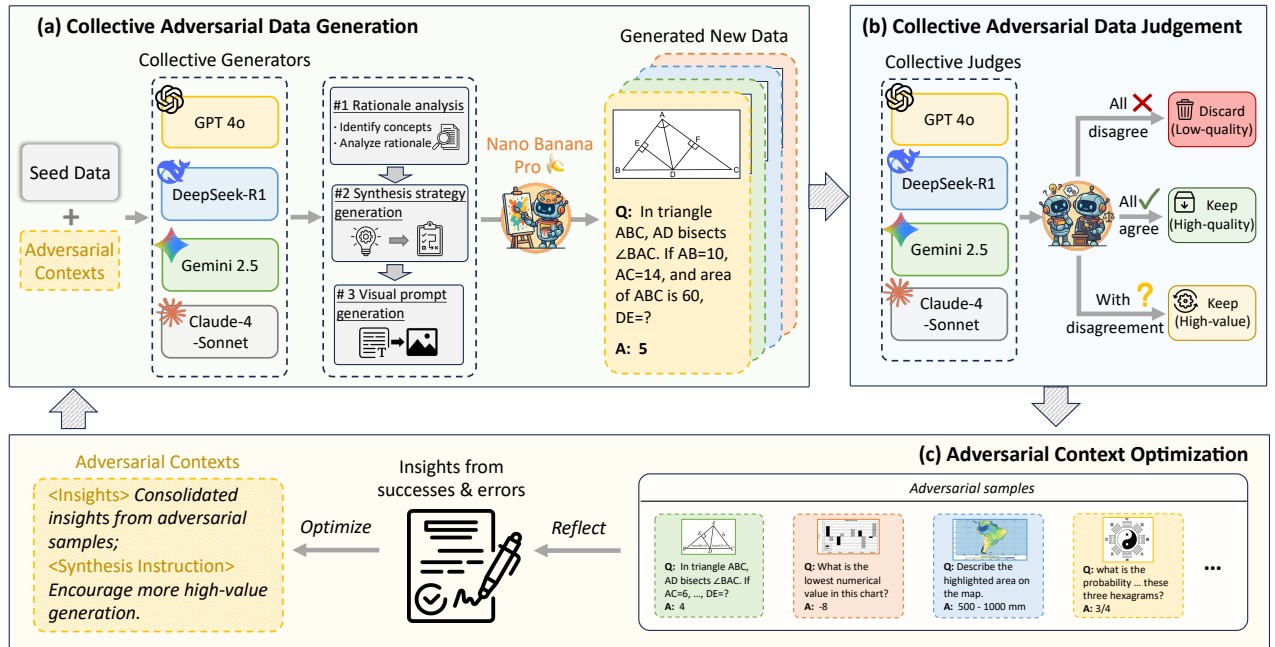

*Figure 2.* **Overview of our proposed Collective Adversarial Data Synthesis (CADS).** CADS operates with two cyclic phases, including (a) Collective Adversarial Data Generation (CAD-Generate), which leverages collective knowledge to generate diverse data and (b) Collective Adversarial Data Judgment (CAD-Judge), which collaboratively assesses the quality of the synthesized data. During these two phases, CADS introduces an (c) Adversarial Context Optimization mechanism to optimize the generation context and encourage challenging and high-value data generation.

cally alters quantitative parameters or conditions (e.g., dimensions, angles, physical constants);

- *(2) Logic Reversion:* it inverts the causal chain of the seed data by swapping the given conditions and the target objective;

- *(3) Auxiliary Extension:* it introduces an auxiliary element to the scene that reinforces the core concept (e.g., construct an altitude or a median into a geometry question);

- *(4) Isomorphic Scenario Transfer:* it maps the core rationale to a completely different visual scenario that shares the same logical structure (e.g., transfer a "collision conservation" problem from billiard balls to vehicle scenarios).

Depending on the specific knowledge domain and rationale identified in the previous step, CAD-Generate dynamically derives these meta strategies into fine-grained generation strategies specific to each seed data. This adaptive process ensures that the synthesis approach is not limited to a fixed set of rules but is flexibly generated on the fly, significantly enhancing the diversity of the synthesized data.

**Step 3: Visual prompt generation.** This step aims to translate the generated new problem into a detailed and precise visual prompt, which will serve as the input for Nano Banana Pro to synthesize the corresponding visual content.

Specifically, we require the prompt to explicitly define the spatial layout, object attributes, specific data values, and geometric relationships inherent to the problem. This enables Nano Banana Pro to accurately interpret the logical constraints, ensuring that the synthesized visual content ($v'$) is strictly aligned with the textual problem description rather than relying on ambiguous hallucinations.

### 3.2.2. COLLECTIVE ADVERSARIAL DATA JUDGMENT

Collective Adversarial Data Judgment (CAD-Judge) employs multiple MLLMs as judges to collaboratively assess the quality of synthesized data, enabling to identify and filter out low-quality instances with issues like semantic misalignment or factual errors:

$$\hat{\mathcal{D}}_{syn} = \mathcal{J}(\mathcal{D}_{syn}) \qquad (2)$$

Specifically, for each synthesized instance $(v', q', a')$, we leverage collective judges $\Pi = \{\pi_1, ..., \pi_K\}$ to verify its solvability. Each judge model $\pi_k$ attempts to solve the synthesized question, generating a prediction $p_k = \pi_k(v', q')$. We then quantify the quality of the instance by calculating a consensus score $C$, which represents the number of models that successfully match the generated ground truth $a'$:

$$C = \sum_{k=1}^{K} \mathbb{I}(p_k = a'), \qquad (3)$$

*Table 2.* **Main Results.** To examine the effectiveness of our synthesized data (i.e., MMSynthetic-20K) and the trained model (i.e., R1-SyntheticVL), we comprehensively benchmark our R1-SyntheticVL with various state-of-the-arts, including general and reasoning-based MLLMs. † denotes evaluation on official weights using VLMEvalKit.

| Model | MathVista | MathVerse | MathVision | MMMU | MMMU-Pro | | CharXiv | | Avg |
|---|---|---|---|---|---|---|---|---|---|
| | | | | | Std-10 | Vision | Reas. | Desc. | |
| *Closed-Source Model* | | | | | | | | | |
| GPT-4o (OpenAI, 2024) | 63.8 | 50.2 | 30.4 | 70.7 | 54.0 | 49.7 | 47.1 | 84.5 | 56.3 |
| Claude-3.5-Sonnet (Anthropic, 2024) | 65.3 | – | 38.0 | 68.3 | 55.0 | 48.0 | 60.2 | 84.3 | – |
| Kimi k1.5 (Kimi et al., 2025) | 74.9 | – | 38.6 | 70.0 | – | – | – | – | – |
| *Open-Source Model Trained using Real Data* | | | | | | | | | |
| MiniCPM-V-2.6-8B (Yao et al., 2024c) | 60.6 | – | – | 49.8 | – | – | – | – | – |
| LLaVA-OV-7B (Li et al., 2024a) | 63.2 | – | – | 48.8 | 29.5 | 18.7 | 23.6 | 48.7 | – |
| LLaVA-OV-1.5-8B (An et al., 2025) | 69.6 | – | 25.6 | 55.4 | 37.4 | 25.2 | – | – | – |
| Mulberry-7B (Yao et al., 2024a) | 63.1 | – | – | 55.0 | – | – | – | – | – |
| R1-VL-7B (Zhang et al., 2025a) | 63.5 | 40.0 | 24.7 | – | – | – | – | – | – |
| Qwen2.5-VL-7B (Bai et al., 2025) | 68.2 | 49.2 | 25.1 | 51.9† | 38.0† | 35.8† | 42.5 | 73.9 | 48.1 |
| R1-Onevision-7B (Yang et al., 2025a) | 64.1 | 46.4 | 29.9 | 52.8† | 35.5† | 31.8† | 27.2† | 55.6† | 42.9 |
| MMR1-7B-RL (Leng et al., 2025) | 72.0 | **55.4** | 31.8 | 50.8† | 34.6† | 32.1† | 41.0† | 69.7† | 48.4 |
| MM-Eureka-7B (Meng et al., 2025) | 73.0 | 50.3 | 26.9 | 55.3 | 37.2† | 34.5† | 38.9† | 73.2† | 48.7 |
| OpenVLThinker-7B (Deng et al., 2026) | 72.3 | 50.3 | 25.9 | 52.0† | 39.1† | 34.4† | 39.0† | 69.7† | 49.1 |
| Vision-R1-7B (Huang et al., 2025c) | 73.5 | 52.4 | 29.4 | 54.2 | 36.5† | 34.2† | 36.6† | 64.5† | 47.7 |
| ThinkLite-VL-7B (Wang et al., 2025b) | 75.1 | 52.1 | **32.9** | 55.5 | 37.5† | 35.5† | 38.7† | 75.2† | 50.3 |
| *Open-Source Model Trained using Synthetic Data Only* | | | | | | | | | |
| **R1-SyntheticVL (Ours)** | **75.6** | 51.2 | 29.1 | **56.3** | **42.0** | **38.7** | 47.8 | 75.5 | **52.0** |

where $\mathbb{I}(\cdot)$ is the indicator function. If $C = 0$, it indicates that none of the models could solve the problem, suggesting potential flaws in the synthesis (e.g., ambiguity or errors). Consequently, we filter out these unsolvable instances to ensure the reliability of the final dataset.

**Adversarial Context Optimization.** During the generation and judgment cycle, CADS introduces an Adversarial Context Optimization mechanism to dynamically refine the data synthesis heuristics. We focus on identifying *high-value adversarial instances* that lie on the decision boundaries of current models. Formally, utilizing the consensus score $C$ derived in Eq. 3, we define the set of adversarial instances $\mathcal{D}_{adv}$ as those that exhibit inter-model disagreement:

$$\mathcal{D}_{adv} = \{(v', q', a') \in \mathcal{D}_{syn} \mid 1 \leq C < K\}. \quad (4)$$

Unlike high-confidence cases where a unanimous consensus is reached ($C = K$) or the filtered unsolvable noise ($C = 0$), these inconsistent samples represent complex problem types where the collective fails to agree despite the problem being solvable.

By exploiting these high-value adversarial instances, we optimize the synthesis strategy via *(1) Reflect:* CADS distills consolidated insights from the disagreements among collective judges, analyzing the underlying patterns of suc-

cesses and errors; *(2) Optimize:* These insights are incorporated into the generation context to dynamically refine the synthesis strategies, thereby progressively encouraging the generation of more high-value and challenging data.

Using CADS, we construct the final multimodal training data MMSynthetic-20K, comprising 20K high-quality synthesized multimodal instances with visual input, a textual instruction and an answer for each question.

## 4. Experiments

### 4.1. Implementation Details

For data synthesis, we adopt a group of four models, including GPT-4o (OpenAI, 2024), Gemini-2.5-Flash (Deep-Mind, 2025a), DeepSeek-R1 (Guo et al., 2025) and Claude-4-Sonnet (Anthropic, 2025), to collectively perform data generation and data judgment. For each seed data, we set the maximum generation iteration as 10. For model training, we adopt Qwen2.5-VL-7B (Bai et al., 2025) as the base model, and perform GRPO (Shao et al., 2024) to train our model. Model optimization is carried out using EasyR1 (Yaowei et al., 2025) codebase, with training conducted on 8 NVIDIA H20 GPUs. For the rollout parameter, we set the number of samples per question to 8. For rein-

*Table 3.* **Ablation study of our proposed Collective Adversarial Data Synthesis (CADS).**

| Nano Banana Pro | Collective Adversarial Data Synthesis | | | MathVista |
| | CAD-Generate | CAD-Judge | Adversarial Context Optimization | |
|---|---|---|---|---|
| | | | | 68.2 |
| ✓ | | | | 70.8 |
| ✓ | ✓ | | | 73.0 |
| ✓ | ✓ | ✓ | | 74.6 |
| ✓ | ✓ | ✓ | ✓ | **75.6** |

forcement learning hyperparameters, we use a global batch size of 128, a rollout batch size of 256, a rollout temperature of 1.0, KL penalty coefficient $1e^{-2}$ and a learning rate of $1e^{-6}$. For sampling policy, we set top p = 1 and top k = -1. The reward function is defined as a combination of accuracy reward and format reward.

### 4.2. Main Experimental Results

To examine the effectiveness of the synthesized data (i.e., MMSynthetic-20K) and the trained model (i.e., R1-SyntheticVL), we conduct extensive experiments and comprehensively benchmark our R1-SyntheticVL with various state-of-the-arts, including general and reasoning-based MLLMs. The evaluation is performed on 6 widely used and challenging datasets, covering the fields ranging from general and mathematical reasoning to chart understanding, and multidisciplinary understanding and reasoning, as shown in Table 2. A detailed description of the benchmarks can be found in the appendix. We compare R1-SyntheticVL with existing state-of-the-art open-source MLLMs trained on real-world data, including general models such as LLaVA-OV-7B (Li et al., 2024a) and MiniCPM-V-2.6-8B (Yao et al., 2024c) and recent reasoning models such as ThinkLite-VL-7B (Wang et al., 2025b), Vision-R1-7B (Huang et al., 2025c), and MMR1-7B-RL (Leng et al., 2025). As shown in Table 2, R1-SyntheticVL achieves the best performance on the majority of benchmarks. For example, on the reasoning-intensive MathVista (Lu et al., 2023) benchmark, our model surpasses ThinkLite-VL-7B and Vision-R1-7B. Notably, on the highly challenging MMMU-Pro (Yue et al., 2025) benchmark, R1-SyntheticVL performs the best, significantly outperforming existing state-of-the-art models. These results show the effectiveness of our proposed CADS in generating high-quality, diverse and challenging multimodal dataset and our synthesized MMSynthetic-20K is capable of providing a valuable resource for mitigating data scarcity and advancing research in MLLM training.

### 4.3. Ablation Study

To investigate the contribution of each component within our CADS, we conduct an ablation study on MathVista bench-

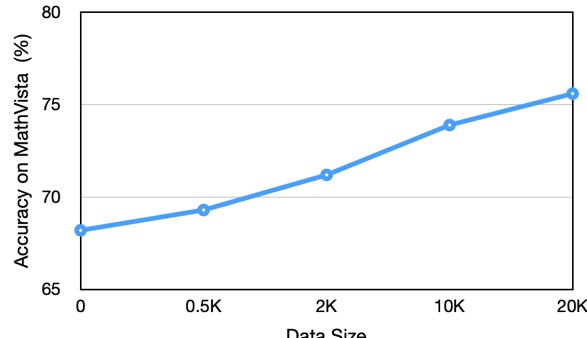

*Figure 3.* **Scaling analysis of synthetic data.**

*Table 4.* **Comparison and complementarity with real data.**

| Training Data | Data Size | MathVista |
|---|---|---|
| Real Data | 2K | 72.2 |
| MMSynthetic (Ours) | 2K | 73.3 |
| Real Data + Ours | 4K | 74.6 |

mark. The results are presented in Table 3. The first row of Table 3 shows the results of the base model, and 'Directly Applying' Nano Banana Pro yields marginal performance improvement as shown in the second row. By introducing our CAD-Generate and CAD-Judge, we can observe a substantial enhancement in performance, which demonstrates that leveraging collective knowledge to ensure diverse generation and filter out low-quality instances largely improves the quality of the synthesized data. Finally, further incorporating the Adversarial Context Optimization yields the best performance, indicating that CADS effectively introduces useful information for model improvement.

### 4.4. Discussion

**Comparison with Real Data.** To further verify the quality of our synthesized data, we conduct a comparison with high-quality real-world data. Specifically, we randomly sample 2,000 instances each from the open-source MM-Eureka dataset (Meng et al., 2025) and our MMSynthetic

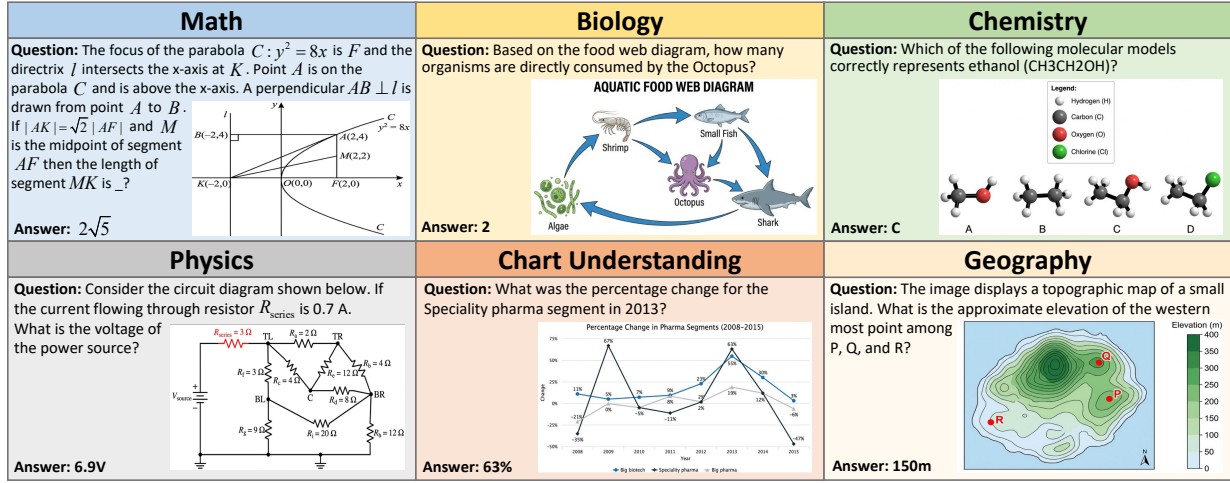

*Figure 4.* **Qualitative illustrations of synthesized samples from MMSynthetic-20K.**

dataset for a fair comparison. We perform GRPO with the same configurations to train the models and evaluate their performance on the MathVista benchmark. As shown in the first two rows in Table 4, the model trained on our synthetic data achieves superior performance, which demonstrates the high quality of our synthesized data, indicating that it can serve as a reliable alternative to real-world data and effectively mitigate the data scarcity issue.

**Complementarity with Real Data.** We further investigate whether our synthesized data could serve as a complementary source to real-world data. We combined the real data with the synthesized data to form a mixed dataset, and train the model on it. As shown in the last row of Table 4, the model trained on this mixture performs the best. This result indicates that our synthesized data introduces diverse reasoning patterns and scenarios that are not fully covered by real-world datasets, thereby further enhancing the model's performance and generalization capability.

**Synthetic Data Scaling Analysis.** We conduct a scaling experiment to investigate the impact of synthesized data size on MLLM performance. Specifically, we train the model using varying amounts of synthesized data, ranging from 0.5k, 2k, 10k, to 20k instances, and evaluate its performance on the MathVista benchmark. As shown in Fig. 3, the model performance improves steadily as the data size increases. After incorporating 20k synthesized samples, the accuracy improves by 7.4% compared to the baseline, demonstrating the effectiveness of our synthesized data. In addition, increasing the data size from 10k to 20k still results in a consistent performance gain, indicating that the model has not yet reached saturation. These results suggest that our synthesized data remains beneficial at larger scales and can provide increasingly diverse reasoning patterns for model training. In this way, as the data scale further expands, our high-quality synthetic data can continuously enrich the

training distribution and progressively enhance the reasoning capability of the model.

**Qualitative Illustrations of Synthesized Data.** Fig. 4 presents samples from MMSynthetic-20K, spanning distinct disciplines including Mathematics, Biology, Chemistry, Physics, and Chart Understanding, etc. As illustrated, the synthesized data exhibit strict visual-semantic alignment, where the generated visual content (e.g., geometric solids, circuit diagrams, and topographic maps) precisely reflects the specific constraints of the textual queries. In addition, these samples require complex multi-step reasoning such as spatial calculation and logical deduction, demonstrating the capability of our proposed CADS to generate high-value and challenging multimodal training data.

## 5. Conclusion

In this paper, we explore data synthesis for Multimodal Large Language Models (MLLMs) and propose Collective Adversarial Data Synthesis (CADS), a novel and general approach to synthesize high-quality, diverse and challenging multimodal data for MLLMs. Specifically, CADS consists of two cyclic phases: Collective Adversarial Data Generation (CAD-Generate), which leverages collective knowledge to generate diverse data, and Collective Adversarial Data Judgment (CAD-Judge), which collaboratively assesses the quality of the synthesized data. In addition, CADS introduces an Adversarial Context Optimization mechanism to optimize the generation context and encourage challenging and high-value data generation. With the proposed CADS, we construct MMSynthetic-20K and train our model R1-SyntheticVL, which demonstrates superior performance on various benchmarks. We hope that our proposed CADS along with MMSynthetic-20K and R1-SyntheticVL will provide valuable resources and offer new insights for addressing the data scarcity issue in MLLM development.

## Impact Statement

This paper presents work whose goal is to advance the field of Machine Learning, specifically by addressing the critical challenge of data scarcity in multimodal large language models (MLLMs). The primary impact of our proposed CADS framework is improving the cost-efficiency and scalability of high-quality data acquisition. By automating the synthesis of complex multimodal data, our method significantly reduces the dependency on expensive and labor-intensive manual data collection and annotation. This offers a sustainable and resource-efficient pathway to train high-performing MLLMs, enabling the community to overcome the limitations of data exhaustion and high annotation costs.

## Acknowledgment

This work was supported by Tencent WeChat Rhino-Bird Focused Research Program WXG-FR-2026-03, and PolyU Internal Fund.

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

# Appendix

## A. Benchmarks

We evaluate our models on the following benchmarks.

- **MathVista** (Lu et al., 2023) serves as a benchmark for assessing the mathematical reasoning and problem-solving capabilities of multimodal large language models. It consists of 6,141 questions spanning diverse mathematical domains, including arithmetic, geometry, algebra, and statistics.

- **MathVerse** (Zhang et al., 2024b) provides a multimodal mathematics evaluation set with 2,612 visual mathematics problems. The dataset covers three major question formats and twelve fine-grained categories, such as plane geometry, solid geometry, and functional analysis.

- **MathVision** (Wang et al., 2024a) comprises 3,040 carefully curated mathematics problems, each paired with corresponding visual information, and is primarily derived from real-world mathematical competitions.

- **MMMU** (Yue et al., 2023) is a comprehensive multimodal benchmark designed to evaluate interdisciplinary understanding and reasoning. It contains approximately 11.5K questions collected from academic sources, including university-level examinations, quizzes, and textbooks.

- **MMMU-Pro** (Yue et al., 2025) is an extended version of MMMU that introduces more challenging evaluation settings. It increases the number of answer options from 4 to 10 and includes a vision-only subset where questions are embedded directly in images. In this work, we evaluate models on the Standard (10-choice) and Vision subsets of MMMU-Pro.

- **CharXiv** (Wang et al., 2024b) is a chart understanding benchmark consisting of 2,323 charts collected from scientific papers. It evaluates both *descriptive* questions about chart elements and *reasoning* questions that require integrating information across visual components.

## B. More Discussions

### B.1. Comparison with other data synthesis methods for MLLMs

We note that there are few studies explore data synthesis for MLLMs. However, existing attempts (Yang et al., 2025b; Huang et al., 2025b; Shi et al., 2025; Linger et al., 2025; Zhang et al., 2025b; Zhao et al., 2024) generally focus solely on synthesizing the textual modality for existing images, or rely on programmatic engines (e.g., Python plotting package) to generate specific visual structures (e.g., charts). Different from previous works, we explore generative models (i.e., Nano Banana Pro (DeepMind, 2025b)) for multimodal data synthesis (i.e., synthesizing both the visual and textual content) and propose a general approach for generating high-quality, diverse and challenging data, effectively enhancing the capabilities of MLLMs in tackling various complex real-world tasks (e.g., general understanding, math reasoning, chart understanding and multidisciplinary understanding). In addition, we provide quantitative comparison with existing data synthesis methods. As shown in table 5, our R1-SyntheticVL achieves the best performance consistently across different benchmarks, further demonstrating that our proposed CADS framework is highly generalizable, producing synthesized data with superior quality and diversity.

*Table 5.* **Comparison with existing data synthesis methods for MLLM.**

| Model | MathVista | CharXiv (Reas.) | CharXiv (Desc.) | MMMU-Pro (Std-10) | MMBench | MME |
|---|---|---|---|---|---|---|
| ECD (Yang et al., 2025b) | - | 40.2 | 74.2 | - | - | - |
| TR-CoT (Linger et al., 2025) | 74.5 | - | - | - | - | - |
| Oasis (Zhang et al., 2025b) | - | - | - | 29.5 | 77.4 | 2037 |
| Genixer (Zhao et al., 2024) | - | - | - | - | 65.3 | 1503 |
| **R1-SyntheticVL (Ours)** | **75.6** | **47.8** | **75.5** | **42.0** | **81.6** | **2344** |

## B.2. Ablation on judge models

We conduct a new ablation study replacing proprietary judges with open-source alternatives (DeepSeek, Qwen2.5-VL-72B, Qwen2.5-VL-7B, and Llama3.2-11B). As shown in Table 6, despite an expected slight performance drop, the model trained on this new data still clearly outperforms the baseline. This demonstrates that our CADS framework is highly robust and not strictly bound to expensive APIs, offering a cost-effective alternative for scalable data generation.

*Table 6.* Ablation on judge models.

|  | Baseline | CADS | CADS with open-source judges |
| --- | --- | --- | --- |
| MathVista | 68.2 | 75.6 | 74.5 |

## B.3. Generalizability across different generators

We conduct experiments by applying CADS to other base generators, i.e., Stable Diffusion and Nano Banana. The new results in table 7 show that integrating CADS improves performance compared to directly using these base models, validating its robust generalizability across diverse generators.

*Table 7.* Generalizability across different generators.

|  | Stable Diffusion (Direct apply) | Stable Diffusion (CADS) | Nano Banana (Direct apply) | Nano Banana (CADS) |
| --- | --- | --- | --- | --- |
| MathVista | 66.3 | 67.8 | 67.9 | 69.5 |

## B.4. Experiments on different base MLLMs

We conduct experiments by applying our MMSynthetic-20K dataset to two additional base models: InternVL3-8B and Qwen2-VL-7B. The new experimental results in Table 8 demonstrate consistent and significant performance gains for both models. This effectively validates the high quality and effectiveness of MMSynthetic-20K in generalizing well across different model families.

*Table 8.* Experiments on different base MLLMs.

|  | InternVL3-8B | +Ours | Qwen2-VL-7B | +Ours |
| --- | --- | --- | --- | --- |
| MathVista | 74.2 | 78.5 | 58.2 | 64.7 |

## B.5. Experiments on supervised finetuning (SFT)

We conduct an SFT experiment using the same MMSynthetic-20K dataset on the Qwen2.5-VL-7B base model. As shown in Table 9, SFT with MMSynthetic-20K improves the MathVista accuracy from 68.2 to 73.4, demonstrating the effectiveness of our constructed dataset.

*Table 9.* Experiments on supervised finetuning (SFT).

|  | Baseline | SFT with MMSynthetic-20K |
| --- | --- | --- |
| MathVista | 68.2 | 73.4 |

## B.6. More analysis of "Seed data"

Here we provide more discussion on the "Seed Data" used in our proposed CADS. (1) Seed selection: We sample initial seeds across a wide range of disciplines from high-quality datasets (i.e., R1-ShareVL-52K (Yao et al., 2025a)) to ensure a broad foundation for synthesis. (2) Impact on Exploration Capability: Seed diversity largely affects the exploration capacity of the CAD-Generate phase. Since the generation relies on the core rationale and domain information extracted from the seeds, a homogeneous seed pool would naturally limit the generated semantic space. (3) We conduct an experiment

removing seeds from specific domains (e.g., Math, Biology). As shown in Table 10, this causes a clear performance drop, confirming that a diverse seed pool is crucial for maximizing the framework's exploratory capabilities.

*Table 10.* "Seed Data" analysis.

|            | CADS | w/o Math | w/o Biology |
|------------|------|----------|-------------|
| MathVista  | 75.6 | 73.2     | 75.1        |

### B.7. Domain composition of MMSynthetic-20k

Our dataset covers 8 major domains: Logic Reasoning (58.89%), Math (28.88%), Chart Understanding (6.82%), Visual Counting (3.07%), Physics (1.02%), Biology (0.66%), Geography (0.35%), and Chemistry (0.32%).

### B.8. Error types

We manually review a random sample of 500 instances filtered out by CAD-Judge. The error distribution is as follows:

(1) Question Errors (43%): Flaws or logical inconsistencies within the generated question;

(2) Ground-Truth Errors (39.6%): Incorrect final answers despite image-text alignment.

(3) Spatial/Geometric Contradictions (13.8%): Visual content contradicts the text (e.g., unclosed shapes);

(4) Missing Crucial Information (2.6%): Lacking necessary visual evidence (e.g., missing axis labels);

(5) Valid but Unsolvable (0.8%): Correct problems that judges failed to solve;

(6) Text Rendering Errors (0.2%): Unreadable text embedded within the image.

This confirms that the vast majority of filtered data are genuine errors rather than merely difficult problems, proving our filtering mechanism is highly precise and necessary.

### B.9. Cost for data synthesis

As previously discussed, AI is running out of data (Jones, 2024), especially for domain-specific data that are inherently scarce and hard to obtain. Even the raw data is available, annotating large-scale multimodal data is prohibitively expensive and time-consuming (Yao et al., 2024a; Xu et al.). To mitigate these constraints, our CADS framework offers an efficient and effective automated solution, which synthesizes high-quality, diverse and challenging multimodal data for MLLMs. Empirically, synthesizing one high-quality instance requires approximately **40 seconds** and incurs an API cost of around **$0.30**. This is substantially more cost-effective than manual data collection and annotation, avoiding the high expense of expert labor and the lengthy time required for data collection, annotation and verification. Furthermore, our generative approach largely eliminates the need to physically collect raw images, which is often difficult in rare or sensitive fields. Looking forward, the scalability of our method can be improved further. As more powerful open-source models emerge, the reliance on commercial APIs may diminish, making CADS an increasingly sustainable and cost-effective pathway for large-scale data synthesis.

## C. More qualitative Illustrations of Synthesized Data

Here we provide more samples from MMSynthetic-20K. As shown in Fig. 5, our CADS is capable of generating high-quality, diverse and challenging multimodal data.

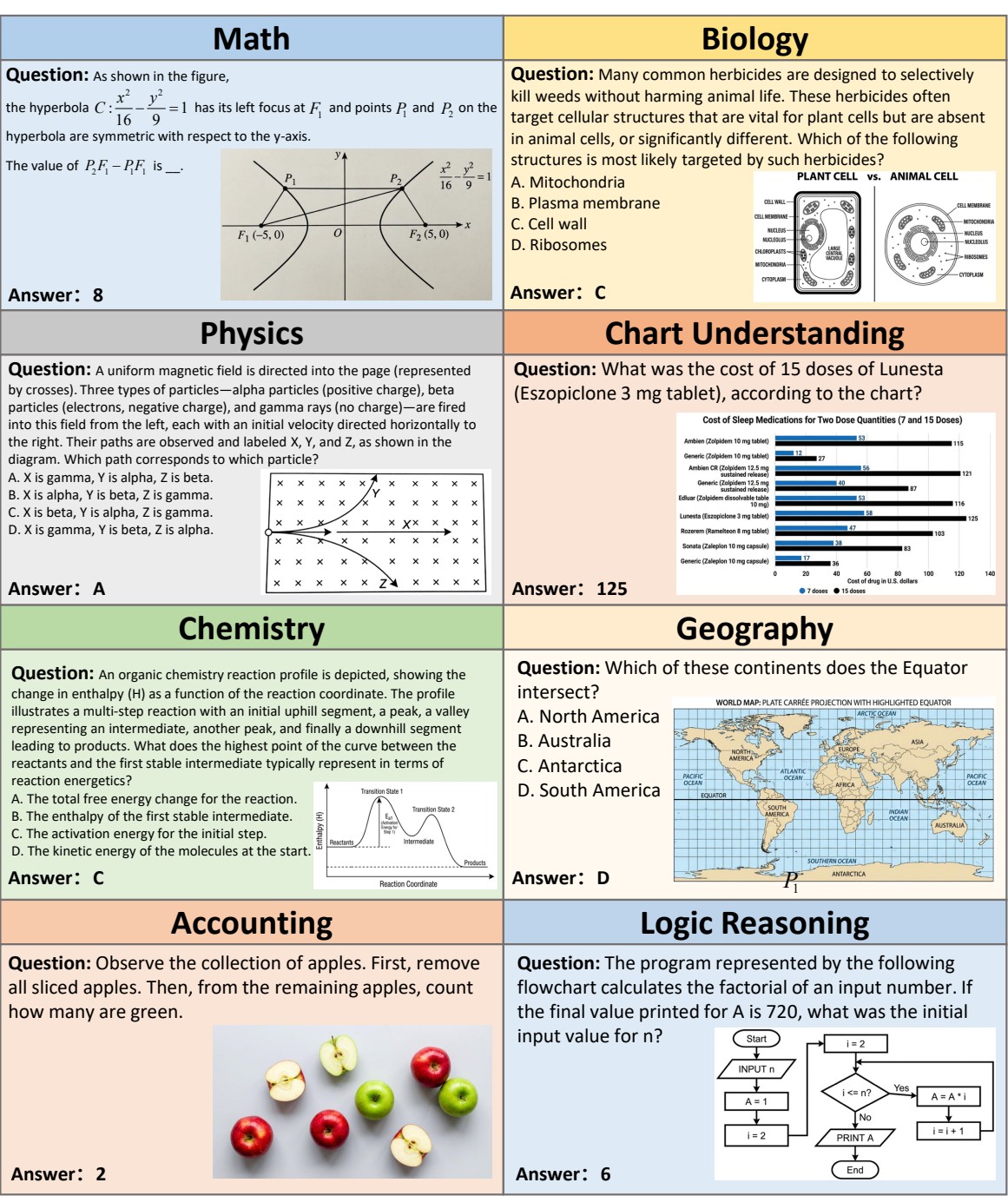

*Figure 5.* **More qualitative illustrations of synthesized samples from MMSynthetic-20K.**

