# OpenReview forum: "R1-SyntheticVL: Is Synthetic Data from Generative Models Ready for Multimodal Large Language Model?"
_ICML.cc/2026/Conference — ICML 2026 regular_

### Official Review · Reviewer_AK2D · 2026-03-11

**Soundness:** 2
**Presentation:** 3
**Significance:** 2
**Originality:** 2
**Overall Recommendation:** 4
**Confidence:** 3

**Summary:**

This paper addresses the scarcity of multimodal training data for MLLMs by proposing a synthetic data generation framework named Collective Adversarial Data Synthesis (CADS).
CADS is designed as an iterative three-stage pipeline:
- CAD-Generate: Multiple MLLMs collaboratively generate problem–answer pairs along with image prompts (i.e., instructions for image generation) based on a given seed.
- CAD-Judge: Multiple MLLM judges attempt to solve the generated samples, and quality is determined via consensus-based filtering (to detect errors and improve reliability).
- Adversarial Context Optimization (ACO): Disagreement samples are used to refine generation contexts and strategies, with the goal of making the outputs more challenging, diverse, and high-quality.
Using this pipeline, the authors construct MMSynthetic-20K, a 20K synthetic multimodal dataset. Based on this dataset, they fine-tune Qwen2.5-VL-7B with GRPO and introduce R1-SyntheticVL, reporting improved reasoning performance across several benchmarks.

**Compliance With Llm Reviewing Policy:**

Affirmed.

**Final Justification:**

The clarifications provided for [W1] and [W2] are clear and effectively address my concerns, particularly regarding the methodological transparency and the robustness of the consensus mechanism.

For W3, the explanation of seed data construction and the effort to prevent benchmark contamination are helpful. It would further strengthen the paper if more concrete details were provided on how the isolation from evaluation benchmarks is strictly guaranteed.

**Key Questions For Authors:**

[Q1] Clarification of Table 1 setup
For the comparison among Stable Diffusion / Nano Banana / Nano Banana Pro / CADS (Ours):
Is the same base model used with only the training dataset changed?
Also, could you clarify how the data synthesis strategies differ across models (e.g., in terms of the Q/A generation pipeline and filtering/judgment mechanisms)?

[Q2] Cost and time breakdown
The reported “40 seconds / $0.30 per instance” requires clarification:
What is the average number of iterations?
How many model calls are involved (image generation, text generation, judgment)?
Is this single-instance latency or parallel throughput?
A comparison with prior synthetic approaches (text-only synthesis, programmatic generation) would improve fairness.

[Q3] Seed provenance and duplication control
What is the exact source of seeds?
How is benchmark contamination prevented (e.g., hashing, embedding-based deduplication, search-based filtering)?

[Q4] Human evaluation
Is there any human validation for image-text alignment, label correctness, or difficulty?
Are there estimates of label noise?

[Q5] Interpretation of “judge failure”
Is “judge fails to solve” ever treated as a proxy for incorrectness?
How are “too difficult” samples distinguished from “erroneous” ones?

**Limitations:**

The paper need to consider the potential biases and risks that may arise from the synthetic data generation process as a limitation.

**Strengths And Weaknesses:**

Strong points

[S1] Novel multimodal synthetic data framework:
The paper presents one of the first structured frameworks that synthesizes fully multimodal training data (including generated images) using generative models. Unlike prior synthetic approaches that rely on captioning existing images or programmatic generation (e.g., Python plotting), this work synthesizes multimodal samples via a generative image model (Nano Banana Pro), which represents a clear conceptual distinction.

[S2] Well-motivated method design addressing quality/diversity/difficulty:
Rather than directly applying image generation models, the paper explicitly identifies limitations in naive synthetic generation—namely quality, diversity, and difficulty—and proposes a structured solution via collective generation, collective judgment, and adversarial context optimization. In particular, the use of multiple generation/judgment models (GPT-4o, Gemini-2.5-Flash, DeepSeek-R1, Claude-4-Sonnet, etc.) appears intended to reduce model-specific bias.

[S3] Practical implications (data cost reduction):
The proposed pipeline potentially reduces reliance on expensive human annotation, copyright-restricted datasets, and manual data curation. The direction toward scalable synthetic multimodal data construction is practically meaningful.

---
Weak points

[W1] Insufficient methodological detail for reproducibility:
- In CAD-Generate, it is not fully clear how collective intelligence is concretely integrated (e.g., how multiple MLLM outputs are aggregated or selected).
- In the ACO stage, especially the “Reflect” component, it is unclear what insights are extracted and how they are incorporated into subsequent context updates.
- Among the four synthesis strategies, it is not clear when and how each strategy is selected (model-autonomous vs rule-based).
- It would be helpful to include qualitative examples showing different strategy outputs from the same seed.
- The rationale for selecting four models (K=4) for consensus is not sufficiently justified.
- The GRPO training setup (hyperparameters, steps, sampling policy, reward definition, etc.) appears under-specified.

[W2] Potential failure modes of consensus-based CAD-Judge:
In domains such as mathematics, geometry, or charts, multiple MLLMs may share the same systematic reasoning errors, allowing incorrect samples to pass consensus filtering. Conversely, discarding samples with C=0 (i.e., none of the judges solve them) may remove genuinely challenging but valid samples, biasing the dataset toward problems solvable by current judges.

[W3] Seed source and contamination concerns:
The seed data is described only as “existing multimodal samples or text descriptions,” without clearly specifying provenance. Given the importance of benchmark contamination control, more explicit clarification on seed source, deduplication policy, and overlap prevention with evaluation benchmarks would strengthen the paper.

[W4] Limited quantitative validation of diversity claims:
While diversity is emphasized conceptually, it is not clearly quantified.
Domain composition, topic distribution, and diversity metrics are not sufficiently analyzed.

---

> ### Author Rebuttal · Authors · 2026-03-31
>
> We sincerely thank the reviewer for the thorough review and helpful feedback. Our detailed responses to each of your concerns are provided below. We will incorporate all new experimental results and clarifications into the final version of the paper.
>
> **W-1: Methodological details.**  Below we clarify the missing details :
> (1) Generators operate independently. Their outputs are not aggregated; instead, each is forwarded individually to the Judge phase. (2) In Reflect, when an adversarial sample causes disagreement among the judge models, we analyze the root cause of this divergence to identify which novel modifications or new knowledge effectively challenged the models. These findings are used as "Consolidated Insights" and are then directly injected into the prompt for next generation. (3) To maximize the generation diversity, we apply every synthesis strategy to every seed problem. (4) We will add qualitative examples to show how a single seed transforms across all four strategies into the revised manuscript. (5) As shown below, while downstream performance improves as K increases, we selected K=4 as the optimal trade-off between performance and computational cost.
> ||1|2|3|4|5|
> |-|-|-|-|-|-|
> |MathVista|73.8|74.1|75.0|75.6|75.8|
>
> (6) Following prior studies, we train the model with learning rate $1\text{e}^{-6}$, global batch size 128,  rollout temperature 1.0 and KL penalty coefficient $1\text{e}^{-2}$. For sampling policy, we set top p=1 and top k = -1. The reward function is defined as a combination of accuracy reward and foramt reward.
>
> **W-2:** (1) CADS mitigates this risk via a heterogeneous ensemble (e.g., GPT-4o, Claude, Gemini). Their distinct architectures and training distributions make identical systematic errors highly unlikely. Furthermore, our manual evaluation of 200 instances shows a very low false-positive rate of 1.5%, proving its practical robustness. (2) While filtering C=0 samples might discard hard problems, our analysis (see **Reviewer 6Bua W-7**) confirms the vast majority are invalid rather than merely difficult. Thus, this filtering is a necessary trade-off for dataset quality.
>
> **W-3 & Q-3: Seed source and contamination concerns.** Our seed data is collected through:
> (1) Sampling from high-quality open-source datasets, i.e., R1-ShareVL-52K [1]. (2) For domains where existing QA pairs are scarce, we use raw textual descriptions (e.g., concept definitions) to prompt the model to generate initial QA pairs. These generated pairs subsequently serve as seed data. Regarding benchmark overlap, we guarantee no evaluation data was used as seeds or prompts. In addition, we implemented a search-based filtering to ensure complete isolation from all test sets.
>
> [1] R1-ShareVL: Incentivizing Reasoning Capability of Multimodal Large Language Models via Share-GRPO
>
> **W-4: Domain composition:** Our dataset covers 8 major domains: Logic Reasoning (58.89%), Math (28.88%), Chart Understanding (6.82%), Visual Counting (3.07%), Physics (1.02%), Biology (0.66%), Geography (0.35%), and Chemistry (0.32%).
>
> **Q-1: Clarification of Table 1 setup.** (1) Yes, the same base model is used with only the training dataset changed. (2) For the baselines (Stable Diffusion, Nano Banana, and Nano Banana Pro), we employed direct generation: the models are simply prompted to expand upon the seed data, without any filtering, judgment mechanisms. In contrast, our CADS  operates Collective Adversarial Data Generation and Collective Adversarial Data Judgment, enabling high-quality generated data and thus better model performance.
>
> **Q-2: Cost breakdown.** (1) On average, the number of iterations is 5.6. Each generation cycle involves 9 model calls: 4 for generation with MLLMs, 4 for judgement with MLLMs, and 1 for image generation. It is parallel throughput.
> (2) Based on available reported results, our generation cost is competitive with prior methods like ECD. Importantly, CADS enables the generation of diverse, complex multimodal data across various domains, offering superior training versatility.
>
> ||API Cost|Computational Cost|
> |-|-|-|
> |ECD|0.2$| 60s|
> |Ours| 0.3$|40s|
>
> **Q-4: Human evaluation.** We randomly sample 200 instances for manual evaluation: (1) Accuracy & Label Noise: 99.5% of the generated samples exhibit accurate image-text alignment, and the overall label noise is exceptionally low at only 2%. (2) Difficulty Distribution: Easy (47%) (requiring basic visual recognition), Medium (38%) (requiring multi-step reasoning), and Hard (15%) (demanding complex, multi-hop logical deductions).
>
> **Q-5: Judge failure.** (1) In CADS, unanimous judge failure (C=0) serves as a heuristic proxy for incorrectness. (2) While distinguishing "too difficult" from "erroneous" autonomously is challenging, our manual analysis (as in Response to **Reviewer 6Bua W-7**) confirms that the vast majority of such failed samples contain fatal flaws rather than being difficult. Thus, this proxy acts as a necessary and highly precise filter.

---

> > ### Author Rebuttal · Reviewer_AK2D · 2026-04-03
> >
> > Thank you for the detailed response. My concerns have been addressed, and I will update my score accordingly.

---

> > > ### Author Response · Authors · 2026-04-06
> > >
> > > We are glad to hear that our rebuttal fully addressed your concerns! Thank you again for your constructive suggestions which made our work more solid and clear. We have incorporated the new experiments and discussions into the revised manuscript.

---

### Official Review · Reviewer_yzqC · 2026-03-13

**Soundness:** 2
**Presentation:** 3
**Significance:** 3
**Originality:** 3
**Overall Recommendation:** 4
**Confidence:** 3

**Summary:**

This paper investigates the feasibility of using advanced generative models (e.g., Nano Banana Pro) to autonomously synthesize high-quality, diverse, and challenging training data for Multimodal Large Language Models (MLLMs). Overall, the submission discusses a central concept of addressing the critical "data exhaustion" crisis in the multimodal domain by moving beyond simple data augmentation towards a sophisticated "Collective Adversarial" synthesis loop. To achieve this,  the authors seek to present a general domain framework called Collective Adversarial Data Synthesis (CADS). CADS leverages the collective intelligence of multiple MLLMs to generate and judge data, while using an Adversarial Context Optimization mechanism to target high-value samples. The authors demonstrate the effectiveness of their method by releasing the MMSynthetic-20K dataset and the R1-SyntheticVL model, which achieves competitive results on reasoning-heavy benchmarks.

**Compliance With Llm Reviewing Policy:**

Affirmed.

**Key Questions For Authors:**

1.  **Ablation on Judge Models:** How does the quality of the synthesized data change if the "Collective Judges" consist of smaller, open-source models rather than proprietary ones?
2.  **Saturation Point:** Figure 3 shows a steady improvement up to 20K. Do the authors have insights into when "diminishing returns" or "model collapse" might occur as the synthetic dataset scales further (e.g., to 100K+)?
3.  **Seed Diversity:** How were the initial "Seed Data" samples selected? Does the diversity of the seeds significantly impact the "exploration" capability of the CAD-Generate phase?

**Limitations:**

The authors have adequately discussed the limitations.

**Strengths And Weaknesses:**

## Strengths
*   **Methodological Rigor:** The CADS framework is well-structured. By integrating "Collective Generation" and "Collective Judgment," it effectively addresses the common pitfalls of synthetic data, such as factual inconsistency and semantic misalignment.
*   **Focus on Difficulty:** The introduction of "Adversarial Context Optimization" is highly commendable. Instead of generating "easy" samples, the framework actively seeks "high-value" instances where models disagree, which is crucial for driving actual model improvement.
*   **Strong Empirical Evidence:** The R1-SyntheticVL model, trained purely on synthetic data, performs remarkably well on MathVista (75.6) and MMMU-Pro, surpassing several state-of-the-art models trained on human-annotated data.
*   **Valuable Resource:** The release of the MMSynthetic-20K dataset provides a high-quality benchmark for the community to explore the potential of purely synthetic multimodal training.

---

## Weaknesses
*   **Cost and Accessibility:** While the authors argue that the 0.30 dollars per instance cost is lower than human experts, the reliance on top-tier commercial APIs (GPT-4o, Claude-4-Sonnet, etc.) still poses a significant financial barrier ($6,000 for 20K samples). This may limit the scalability and reproducibility for researchers without substantial funding.
*   **Base Generator Bias:** The framework’s success is heavily dependent on the "Nano Banana Pro" generator. The paper lacks a deep discussion on how the system would handle inherent biases or systematic failures if the base generator has specific "blind spots" in certain visual domains (e.g., specialized medical or rare industrial scenes).
*   **Semantic Drift Potential:** In the "Collective Adversarial" loop, there is a risk of models converging on a "synthetic-style" logic that satisfies the judge models but may diverge from real-world human reasoning. More analysis on the "naturalness" of the reasoning chains would strengthen the paper.

---

> ### Author Rebuttal · Authors · 2026-03-31
>
> We highly appreciate the reviewer's valuable comments and constructive suggestions. Please find our point-by-point responses to your questions below. All newly conducted experiments and additional discussions will be included in the revised manuscript.
>
> **W-1: Cost and Accessibility.** First, while API costs exist, $0.30/instance is still much cheaper than hiring human experts for complex reasoning annotation, offering a scalable solution to the multimodal data scarcity problem.
> Second, to ensure accessibility for researchers with less funding, we conduct a new study using open-source models as the judge models. As shown in the table in the response to **Q-1: Ablation on Judge Models**, this open-source setup still significantly improves baseline performance.
> Third, our CADS is the first work to explore synthesizing multimodal data using generative models. As discussed in Section B.2, the scalability and cost-efficiency of our framework will continue to improve over time. As more powerful open-source models emerge, our reliance on commercial APIs will naturally diminish in the future.
>
>
>
> **W-2: Base Generator Bias.**  Here we provide the discussion on this issue.
> First, the current base generator, Nano Banana Pro, already demonstrates strong generative capabilities across most domains. As in Figs. 4 and 5, our CADS successfully synthesizes high-fidelity and complex visual content for various disciplines.
> Second, we acknowledge that any generator may have inherent "blind spots" and fail in highly specialized areas. However, our collective judge mechanism effectively mitigates the negative impact caused by these failures by identifying and filtering out low-quality instances, thus preventing the injection of erroneous or hallucinated data into the final training set.
> Third, CADS is designed as a model-agnostic framework. For highly specialized applications, the base generator can be seamlessly replaced with domain-specific fine-tuned models (e.g., diffusion models for medical imaging).
>
> **W-3: Semantic Drift Potential.** We agree that semantic drift is a critical challenge in autonomous data generation, however, in our CADS framework, this risk is effectively minimized:
> (1) Similar to prior studies, every synthesis iteration in CADS is based on real-world seed data (Section 3.2.1). Since the core concepts and rationales originate from real-world seeds, the underlying logic is forced to stay aligned with natural human reasoning.
> (2) We employ multiple strong judge models rather than a single evaluator. Since these cutting-edge models are already highly aligned with human preferences (e.g., via RLHF), requiring a consensus among them successfully prevents the generator from overfitting to the specific synthetic style of any single model.
> (3) We further examine a random sample of 500 instances, confirming that 99.4% generated data maintain naturalness.
>
> **Q-1: Ablation on Judge Models.** As suggested, we conduct new experiments replacing the proprietary judges with the open-source models (DeepSeek, Qwen2.5-VL-72B, Qwen2.5-VL-7B, and Llama3.2-11B). As shown below, despite an expected slight performance drop, the model trained on this new data still clearly outperforms the baseline. This demonstrates that our CADS framework is robust and not strictly bound to expensive APIs, offering a cost-effective alternative for scalable data generation.
>
> ||Baseline|CADS|CADS with open-source judges|
> |-|-|-|-|
> |MathVista|68.2|75.6|74.5|
>
> **Q-2: Saturation Point.**  Following established scaling laws, model performance typically follows a power-law improvement before plateauing. Given our current scaling curve (Fig. 3), we hypothesize the saturation point for a 7B-parameter model might emerge around the 70K–100K. Furthermore, this saturation largely depends on data diversity and difficulty. We hypothesize that continuously expanding seed diversity and elevating reasoning complexity will effectively delay the saturation. We plan to empirically verify these scaling properties in future work.
>
> **Q-3: Seed Diversity.**  (1) Seed selection:  We sampled initial seeds across a wide range of disciplines from high-quality datasets (i.e., R1-ShareVL-52K [1]) to ensure a broad foundation for synthesis.
> (2) Impact on Exploration Capability: Seed diversity largely affects the exploration capacity of the CAD-Generate phase. Since the generation relies on the core rationale and domain information extracted from the seeds, a homogeneous seed pool would naturally limit the generated semantic space.
> (3) We conduct a new experiment removing seeds from specific domains (e.g., Math, Biology). As shown below, this causes a clear performance drop, confirming that a diverse seed pool is crucial for maximizing the framework's exploratory capabilities.
>
> ||CADS|w/o Math|w/o Biology|
> |-|-|-|-|
> |MathVista|75.6|73.2|75.1|
>
> [1] R1-ShareVL: Incentivizing Reasoning Capability of Multimodal Large Language Models via Share-GRPO

---

> > ### Author Rebuttal · Reviewer_yzqC · 2026-04-04
> >
> > The authors have satisfactorily addressed most of my concerns in the rebuttal. The additional clarifications and results provided further strengthen the paper's contributions. Therefore, I am pleased to maintain my positive rating and support its acceptance.

---

> > > ### Author Response · Authors · 2026-04-06
> > >
> > > We are glad to hear that our rebuttal satisfactorily addressed most of your concerns! Thank you again for your constructive suggestions which made our work more solid and clear. We have incorporated the new experiments and discussions into the revised manuscript.

---

### Official Review · Reviewer_eacb · 2026-03-13

**Soundness:** 4
**Presentation:** 4
**Significance:** 2
**Originality:** 4
**Overall Recommendation:** 4
**Confidence:** 5

**Summary:**

This paper proposes a well-designed synthetic data generation pipeline for MLLMs, termed Collective Adversarial Data Synthesis (CADS). The method is motivated by the limitations of directly use of text-to-image generators leading to low-quality, homogeneous, and naive samples. To address this, the paper introduces three main components:
- **CAD-Generate**, where multiple MLLMs jointly analyze seed examples, derive synthesis strategies, and produce detailed visual prompts for image generation;
- **CAD-Judge**, where multiple MLLMs act as judges to verify the solvability and consistency of synthesized samples; and
- **Adversarial Context Optimization**, which mines disagreement cases among judges and feeds them back into the generation context to encourage more challenging and high-value samples.

Based on this framework, the authors construct MMSynthetic-20K and train R1-SyntheticVL based on Qwen2.5-VL-7B using GRPO. Experiments show superior results, while ablations suggest that each proposed component contributes to the final performance.

**Compliance With Llm Reviewing Policy:**

Affirmed.

**Key Questions For Authors:**

- The constructed data seems suitable for SFT, why authors do not have trials?

**Limitations:**

The impact statement covers cost-efficiency improvement and scalability of multimodal data acquisition. By automating the synthesis of complex multimodal training data, the method may reduce the reliance on expensive manual collection and annotation, and thus provide a more sustainable path for training MLLMs under data scarcity, which is **adequate** for this paper.

**Strengths And Weaknesses:**

# Strengths
- **Self-contained motivation and design.** CADS proposed is well aligned with its motivation. Compared with direct generation, CADS explicitly addresses the three main challenges identified , i.e., quality, diversity, and difficulty. By collective generation, collective judgment, and adversarial context optimization. The empirical results also support this design choice, as CADS substantially outperforms direct generation on MathVista, suggesting that the proposed pipeline indeed produces higher-quality multimodal training data.
- **Potential for scalable deployment.**  The paper provides some evidence of scalability: performance improves steadily from 0.5K to 20K synthetic samples in Fig. 3, with no clear saturation at 20K. The authors also report in Appendix B.2 that synthesizing one high-quality instance takes about 40 seconds and costs around $0.30. However, practical scalability remains a concern given the heavy multi-model pipeline and reliance on commercial APIs.
- **Well-motivated adversarial augmentation.** The motivation behind adversarial context optimization is interesting. Instead of discarding all disagreement cases, the paper treats partially inconsistent samples as high-value instances near the models’ decision boundary, i.e., difficult but still solvable cases. This gives the adversarial component a clearer role beyond heuristic filtering: it actively pushes synthesis toward harder and more informative multimodal data. The ablation also provides some support, as adding this component further improves MathVista from 74.6 to 75.6.
# Werakness
- **Lack of comprehensive benchmark.** The paper mainly claims their data pipeline is more curated for MLLM training, where validation of data quality would require training **at least two different** base MLLMs on the synthesized dataset. The current paper mainly demonstrates gains on a single backbone, Qwen2.5-VL-7B, which limits the evidence that MMSynthetic-20K generalizes across model families.
- **Lack of experiments on SFT.** Since MMSynthetic-20K consists of standard multimodal instruction-answer pairs, it appears equally suitable for SFT. However, the paper provides no controlled SFT baseline, nor any comparison between SFT and GRPO on the same data. Is it a intentional choice? SFT also plays an important role in MLLM training, and additional experiments on SFT would better support the paper’s main claim.

---

> ### Author Rebuttal · Authors · 2026-03-31
>
> We sincerely thank the reviewer for their constructive feedback and insightful suggestions. Below, we address your specific concerns point-by-point. All new experiments and discussions will be fully integrated into the revised manuscript.
>
> **W-1: Experiments on different base MLLMs.** We thank the reviewer for this constructive suggestion. As suggested, we conduct new experiments by applying our MMSynthetic-20K dataset to two additional base models:  InternVL3-8B and Qwen2-VL-7B.
> The new experimental results in the table below demonstrate consistent and significant performance gains for both models. This effectively validates the high quality and effectiveness of MMSynthetic-20K in generalizing well across different model families. We will include these experiments in the revised manuscript.
>
> |                |InternVL3-8B|+Ours | Qwen2-VL-7B |+Ours |
> |-------------|---|-----------------------|--------|----|
> |MathVista|74.2| 78.5 | 58.2|64.7|
>
>
> **W-2: Experiments on SFT.** We appreciate the reviewer's insightful suggestion. Our initial focus was primarily on RL as it has recently emerged as the mainstream paradigm for incentivizing reasoning capabilities in MLLMs. However, we fully agree that SFT also plays an important role in MLLM training. As suggested, we conduct a new SFT experiment using the exact same MMSynthetic-20K dataset on the Qwen2.5-VL-7B base model. As shown in the table below, SFT with our dataset provides a clear performance improvement over the baseline, demonstrating the effectiveness of our constructed data. We will add this experiment to the revised manuscript.
>
> |                |Baseline|SFT with   MMSynthetic-20K                 |
> |-------------|---|-------------------------------|
> |MathVista|68.2|  73.4     |

---

> > ### Author Rebuttal · Reviewer_eacb · 2026-04-02
> >
> > Responses fully resolve my concerns.

---

> > > ### Author Response · Authors · 2026-04-06
> > >
> > > We are glad to hear that our rebuttal fully resolved your questions! Thank you again for your constructive suggestions which made our work more solid and clear. We have incorporated the new experiments and discussions into the revised manuscript.

---

### Official Review · Reviewer_6Bua · 2026-03-13

**Soundness:** 2
**Presentation:** 3
**Significance:** 3
**Originality:** 2
**Overall Recommendation:** 4
**Confidence:** 3

**Summary:**

The paper presents a significant advancement in multimodal data synthesis by introducing a collective approach that successfully mitigates the common issues of low quality and limited diversity in generative data. The empirical results are robust, showing that a model trained purely on synthetic data can outperform several models trained on real-world datasets across diverse reasoning benchmarks.

**Compliance With Llm Reviewing Policy:**

Affirmed.

**Final Justification:**

The detailed response has addressed most of my concerns, therefore I will raise my score accordingly.

**Key Questions For Authors:**

See weakness

**Limitations:**

The authors should explicitly discuss the economic and computational overhead associated with the CADS framework and others.

**Strengths And Weaknesses:**

**Strengths:**
1. The CAD-Generate phase leverages multiple MLLMs (GPT-4o, Gemini 2.5, DeepSeek-R1, and Claude-4-Sonnet) to jointly extract rationales and generate strategies, which effectively breaks the inherent biases of single-model generation.

2. R1-SyntheticVL achieves an average score of 52.0 across major benchmarks, outperforming state-of-the-art open-source models like Qwen2.5-VL-7B (48.1) and ThinkLite-VL-7B (50.3). This demonstrates the high impact of the synthesized data on model reasoning.

3. Scaling analysis in Fig. 3 shows a consistent performance gain as data size increases from 0.5K to 20K, indicating that the model has not yet reached saturation and can benefit from further synthesis.


### Weaknesses

1. In Eq. 3, the consensus score $C$ is defined using an indicator function $\mathbb{I}(p_k = a')$. However, the manuscript does not specify how the equality operator ($=$) is implemented for open-ended or complex reasoning answers, where exact string matching is often insufficient.

2. Eq. 4 defines adversarial instances $\mathcal{D}_{adv}$ as those where $1 \le C < K$. The paper provides no formal explanation or empirical justification for why instances where $C=0$ (unsolvable by all judges) are excluded from the reflection and optimization process, as these might also contain valuable failure mode insights.

3. The methodology relies heavily on an ensemble of four high-end proprietary models as judges (GPT-4o, Claude-4, etc.). The manuscript does not explore how the quality of the dataset might degrade if smaller or open-source models were used for the collective judgment phase.

4. The synthesis time of 40 seconds per instance implies a total generation time of approximately 222 hours for 20K samples. This high latency limits the scalability of the method for ultra-large-scale data generation compared to faster programmatic methods.

5. Table 1 provides a comparison between Nano Banana and Nano Banana Pro, but the paper does not show results of applying the CADS framework to other state-of-the-art generators (e.g., Stable Diffusion 3 or FLUX) to confirm its generalizability across different base models.

6. While the "Adversarial Context Optimization" is credited with the best performance in Table 3, the paper lacks a detailed qualitative breakdown or table showing examples of the actual "Consolidated insights" that were distilled during the "Reflect" step.

7. The manuscript mentions "low data quality" and "multimodal misalignment" as critical bottlenecks for direct generation. However, there is no quantitative analysis showing the distribution of error types (e.g., spatial contradiction vs. factual error) that the CAD-Judge successfully filters out.

---

> ### Author Rebuttal · Authors · 2026-03-31
>
> We deeply appreciate the reviewer's insightful comments and valuable suggestions. Please find our detailed responses to your specific questions and concerns below. We will include the newly-added experiments and discussions into our revised manuscript.
>
> **W-1: Equality operator implementation.** Following prior studies, we adopt LLM judge (GPT-4o) for open-ended or complex reasoning answers when exact string matching fails.
>
> **W-2: Justification for the Adversarial Instance.** We agree that incorporating unsolvable instances could provide valuable insights. However, our new experiment (table below) demonstrates that including them yields only marginal performance gains. This is largely because, compared to negative samples, positive samples can provide more direct and constructive guidance for correct generation. Incorporating too many purely negative samples offers limited help to the model and can reduce overall generation efficiency. We will explore the efficient utilization of negative samples in future work.
>
> ||CADS|CADS with unsolvable instances|
> |-|-|-|
> |MathVista|75.6|75.8|
>
> **W-3: Ablation on judge models.** As suggested, we conduct a new ablation study replacing proprietary judges with open-source alternatives (DeepSeek, Qwen2.5-VL-72B, Qwen2.5-VL-7B, and Llama3.2-11B). As shown below, despite an expected slight performance drop, the model trained on this new data still clearly outperforms the baseline. This demonstrates that our CADS framework is highly robust and not strictly bound to expensive APIs, offering a cost-effective alternative for scalable data generation.
>
> ||Baseline|CADS|CADS with open-source judges|
> |-|-|-|-|
> |MathVista|68.2|75.6|74.5|
>
> **W-4: Generation cost.** We would like to clarify that:
> (1) Our time efficiency is actually comparable to, or even better than, that of programmatic methods. For example, recent programmatic approaches ECD [1] report requiring ~20 seconds for initial generation and an additional ~40 seconds for diversification per image, while our total synthesis time is ~40 seconds per instance.
> (2) More importantly, programmatic engines are inherently limited to specific visual structures (e.g., charts). In contrast, our CADS efficiently generates diverse, complex multimodal data across various domains, offering superior versatility and effectiveness for MLLM training.
> (3) In addition, as the first work to explore synthetic multimodal data from generative models for MLLMs, we will actively seek to further reduce generation latency in our future work.
>
> **W-5: Generalizability across different generators.** As suggested, we conduct new experiments by applying CADS to other base generators, Stable Diffusion and Nano Banana. The new results in the table below show that integrating CADS improves performance compared to directly using these base models,  validating its robust generalizability across diverse generators.
>
> ||Stable Diffusion (Direct apply)|Stable Diffusion (CADS)                   |Nano Banana (Direct apply)|Nano Banana (CADS)|
> |-|-|-|-|-|
> |MathVista|66.3|67.8|67.9|69.5|
>
> **W-6: Consolidated Insights.** In the Reflect step, when an adversarial sample causes disagreement, it investigates what novel modifications or new knowledge were introduced compared to the original seed, and why these specific changes effectively challenge the judge models. These findings are then utilized as the  "Consolidated Insights", which act as strategic prompts that encourage the generator to actively explore and synthesize more problems with these high-value characteristics.
>
> For instance, in a spatial reasoning task, the generator might modify a simple object-counting seed by introducing severe partial occlusion and overlapping elements in the image. When judge models fail, the distilled insight dictates: *"Introducing partial occlusion effectively evaluates robust instance tracking. Actively employ overlapping visual elements to challenge spatial resolution in future generations."* We will include a comprehensive qualitative illustrations show these insights across diverse domains in the revised manuscript.
>
> **W-7:  Error types.** We manually review a random sample of 500 instances filtered out by CAD-Judge. The error distribution is as follows:
>
> (1) Question Errors (43%): Flaws or logical inconsistencies within the generated question;
>
> (2) Ground-Truth Errors (39.6%): Incorrect final answers despite image-text alignment.
>
> (3) Spatial/Geometric Contradictions (13.8%): Visual content contradicts the text (e.g., unclosed shapes);
>
> (4) Missing Crucial Information (2.6%): Lacking necessary visual evidence (e.g., missing axis labels);
>
> (5) Valid but Unsolvable (0.8%): Correct problems that judges failed to solve;
>
> (6) Text Rendering Errors (0.2%): Unreadable text embedded within the image.
>
> This confirms that the vast majority of filtered data are genuine errors rather than merely difficult problems, proving our filtering mechanism is highly precise and necessary.

---

> > ### Author Rebuttal · Reviewer_6Bua · 2026-04-04
> >
> > Thank you for the detailed response. My concerns have been addressed, and I will update my score accordingly.

---

> > > ### Author Response · Authors · 2026-04-06
> > >
> > > We are glad to hear that our rebuttal fully resolved your questions! Thank you again for your constructive suggestions which made our work more solid and clear. We have incorporated the new experiments and discussions into the revised manuscript.

---

### Decision · Program_Chairs · 2026-04-30

**Decision:**

Accept (regular)

**Comment:**

This paper proposes a new approach to synthesize high-quality and diverse multimodal data for MLLMs. The analysis of limitations of generated samples by simply employing advanced generative models is insightful. Motivated by the observed limitation pattern, the proposed CADS framework is designed as two cyclic phases to adversarially generate high-quality samples and collectively employ multiple judges. The authors construct a new dataset, MMSynthetic-20K, using CADS and further train a new model, R1-SyntheticVL, on the constructed MMSynthetic-20k. R1-SyntheticVL exhibits superiority compared with many open-sourced and closed-sourced MLLMs across extensive benchmarks.

The reviewers consistently acknowledge the strengths of the adversarial and collective generation pipeline, potential scalability, and strong empirical results achieved by training on MMSynthetic-20k. The authors provide a more detailed elaboration of CADS, a justification of the generation cost, additional experiments on more MLLM architectures, and complementary ablations, addressing most concerns raised by reviewers.

Regarding all reviewers holding a positive attitude, I recommend weak acceptance of this paper.